# PLANNER AND EXECUTOR :
# COLLABORATION BETWEEN DISCRETE DIFFUSION AND AUTOREGRESSIVE MODELS IN REASONING

## ABSTRACT

Current reasoning models achieve high accuracy but require long token sequences, making them costly. Discrete diffusion language models (DDLMs) offer parallel, flexible generation within a fixed number of tokens. This motivates a hybrid design where a DDLM serves as the planner and an autoregressive model (ARM) as the executor, combining efficiency with accuracy. We conduct a systematic study of such planner–executor pairings across text- and latent-space collaboration. Results show that DDLM→ARM collaboration is most effective, especially when interaction occurs in latent space. A learned projector maps DDLM latents into the ARM's embedding space, bypassing some of diffusion's limitations and enabling substantial gains on challenging reasoning tasks. For instance, when DDLM→ARM communication shifts from text space to latent space, accuracy improves from 27.0% to 54.0% on DART-5 and from 0.0% to 14.0% on AIME24. With only 64 planner tokens and ∼5 executor tokens, the latent-space pipeline surpasses Qwen3.1-7B on DART-5 and AIME while only using $1.9\% - 2.2\%$ tokens,

and performs slightly below DeepSeek-R1 while operating with a similarly negligible token budget and only a 3B-sized executor. These findings highlight that diffusion's global revision and autoregression's finalization are complementary, and that latent exchange enables budget-aware reasoning without sacrificing robustness.

## 1 INTRODUCTION

Over the past few years, autoregressive language models (ARMs) (Bengio et al., 2003) have emerged as the dominant paradigm in natural language processing and artificial intelligence, achieving remarkable performance across a wide range of applications (Achiam et al., 2023b;a; Anthropic, 2023; Team et al., 2023, *inter alia*). These models have achieved substantial progress in reasoning tasks through structured approaches such as step-by-step reasoning (Kojima et al., 2022; Cobbe et al., 2021), chain-of-thought prompting (Wei et al., 2023), and plan-and-solve strategies (Wang et al., 2023). More recently, specialized reasoning models (Xu et al., 2025), such as DeepSeek-R1 (Guo et al., 2025) and Qwen3 (Yang et al., 2025b), have demonstrated state-of-the-art performance on challenging mathematical and logical

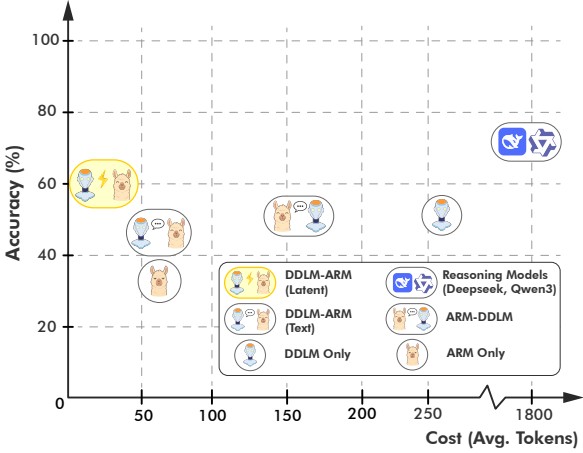

Figure 1: Accuracy–cost trade-offs across planner–executor configurations. DDLM→ARM, particularly with latent-space exchange, achieves higher reasoning accuracy at lower token budgets compared to ARM-only, ARM→DDLM, and SOTA reasoning models.

benchmarks (Wei et al., 2022a; Sun et al., 2025).

In parallel, discrete diffusion language models (DDLMs) (Yu et al., 2025; Li et al., 2025c) have recently attracted growing attention, spurred by findings that they can surpass ARMs in complex reasoning and planning tasks (Ye et al., 2024b). Novel approaches such as Diffusion-of-Thought (Ye et al., 2024c), hybrid strategies that integrate discrete and uniform diffusion (von Rütte et al., 2025; Li et al., 2025a), and reinforcement learning for DDLMs (Zhao et al., 2025) highlight their potential in reasoning tasks.

These two paradigms offer complementary strengths: autoregressive decoding produces fluent, human-comprehensible sequences through next-token prediction, while diffusion enables flexible token generation, a property that recent studies suggest is particularly advantageous for planning tasks. What remains underexplored is how these families could *collaborate* to solve reasoning tasks: Should planning be delegated to DDLMs while execution remains ARMs? Can collaboration happen purely through text, or does exchanging information in latent space unlock different behaviors?

This paper is an *investigation* of DDLM–ARM collaboration for reasoning. We study planner–executor pairings of these language models, and we compare two communication channels: (i) **text space**, where a planner emits a textual plan that conditions the executor, and (ii) **latent space**, where a learned projector maps diffusion representations directly to the executor's embedding space. Figure 1 illustrates the resulting accuracy–cost trade-offs across these configurations, highlighting how latent-space collaboration shifts the efficiency frontier.

Our goal is *not* to propose a single system that "solves long reasoning with fewer tokens." Instead, we treat token budgets as a controlled variable and report token savings and their compute implications as an *outcome* of the collaboration dynamics we uncover.

Our contributions are as follows:

1. **Systematic study of planner–executor roles.** We evaluate four pairings (ARM→ARM, ARM→DDLM, DDLM→ARM, DDLM→DDLM) on diverse reasoning benchmarks, isolating when DDLM should plan ARM should execute.

2. **Two collaboration channels.** We compare *text-space* prompting against a *latent-space* projector that maps diffusion states into the executor's embedding space, quantifying when latent exchange yields superior robustness and fluency.

Together, these results chart when and how DDLMs and ARMs *complement* each other for reasoning, and they surface design principles for future hybrid planners and executors where compute, fluency, and robustness can be balanced explicitly rather than implicitly through longer chains of tokens.

## 2 RELATED WORK

**Autoregressive reasoning.** Autoregressive (AR) language models achieve strong reasoning accuracy, especially with chain-of-thought (CoT) prompting and its variants such as zero-shot CoT and self-consistency (Wei et al., 2022b; Kojima et al., 2022; Wang et al., 2023b). These methods elicit multi-step, token-by-token traces that reliably boost accuracy on arithmetic, commonsense, and symbolic tasks. However, the sequential nature of AR decoding makes long CoT traces computationally expensive at inference time and can lead to verbosity or drift in long-horizon problems (Chowdhery et al., 2022). This motivates exploring non-autoregressive mechanisms that decouple "thinking cost" from output length.

**Discrete diffusion for language.** Discrete/latent diffusion language models generate text by iterative denoising with parallel token updates and flexible order, trading a fixed number of steps for global edits (Austin et al., 2021; Li et al., 2022). Early discrete diffusion underperformed AR LMs on standard LM metrics (e.g., perplexity) (Austin et al., 2021), but subsequent work improved controllability and quality via better objectives and guidance (Li et al., 2022). Conceptually, diffusion offers compute control (fixed steps) and global revision, yet practical challenges remain: weaker surface fluency and alignment compared to AR decoders, especially for long, well-formed ratio-

nales. Recent diffusion-of-thought methods integrate CoT-style reasoning into the diffusion process and report competitive results on multi-step tasks, indicating growing viability of diffusion-based reasoning (Ye et al., 2024a).

**Hybrid planner–executor architectures.** A complementary line decouples *planning* from *execution* in modular agents: a high-level planner proposes a structured plan; an executor realizes it in the environment or produces the final answer (Erdogan et al., 2025). Such factorization improves long-horizon reliability by letting different modules specialize. Our work follows this spirit for *reasoning*: we operationalize a *diffusion LM as the planner* (efficient, globally revisable reasoning trajectory) and an *AR LM as the executor* (fluent, well-aligned finalization).

Prior planner–executor systems overwhelmingly use AR models for both planning and execution; diffusion-based planners for language reasoning remain largely unexplored. To our knowledge, no prior work systematically studies $DDLM \rightarrow ARM$ as a planner–executor pairing for reasoning tasks, nor compares it against ARM→ARM, ARM→DDLM, and DDLM→DDLM baselines under a unified evaluation. Our study fills this gap and further proposes latent-space communication to mitigate the fluency bottleneck when passing diffusion plans in text.

## 3 METHODS

We aim to investigate the potential benefits of fostering collaboration between ARMs and DDLMs through a planner–executor framework. We begin by defining the framework itself, before outlining the two collaboration strategies under consideration.

**Planner–Executor Framework** We use the terms *planner* and *executor* to describe two complementary roles:

**Planner.** We define a *planner* as a language model whose output is intended to support the solution of a reasoning task, without producing the final answer. In a reasoning model, this corresponds to the "thinking" phase, if it were generated separately from the final response. For example, in step-by-step reasoning techniques such as *Chain-of-Thought* (Wei et al., 2023), the planner can be viewed as the component responsible for generating intermediate reasoning steps without the final prediction. More explicitly, in the *Plan-and-Solve* prompting technique (Wang et al., 2023a) which has been shown to improve performance on mathematical reasoning problems, the planner is the module that generates the plan.

**Executor.** The *executor* is a language model responsible for producing the final answer, given the original question and the planner's output, without additional explicit reasoning.

**Communication Channels** We investigate two ways of transmitting information from the planner to the executor. These approaches differ in how they encode, transmit, and interpret the reasoning signal :

**Text-space collaboration.** In the first setup, the planner produces an explicit textual plan, which is then appended directly to the executor's input prompt. This scheme is attractive for its simplicity: it requires no architectural changes, and it mirrors the widely adopted chain-of-thought paradigm, except that plan generation is handled by one model (e.g., DDLM), while a separate model (e.g., ARM) is responsible for finalizing the reasoning into a concrete answer. Another key advantage of text-based collaboration is its full interpretability, as intermediate reasoning steps remain visible and verifiable. However, one may question whether the effectiveness of this approach depends too heavily on the planner's ability to produce fluent text. In fact, diffusion-based models can produce reasoning chains that are logically inconsistent, since their iterative denoising process emphasizes local token updates without guaranteeing global logical coherence. This can lead to higher sequence error rates, as shown by Feng et al. (2025), which may or may not limit their effectiveness as planners in generating prompts for the executor. While we do not claim that this will automatically degrade performance, we were inspired by the Chain of Continuous Thought approach (Hao et al., 2024), which highlights the possibility of reasoning and collaboration occurring in a latent space rather than being strictly tied to surface text. The idea that language models could reason and collaborate freely, without being constrained by linguistic fluency, and only translate their findings into text

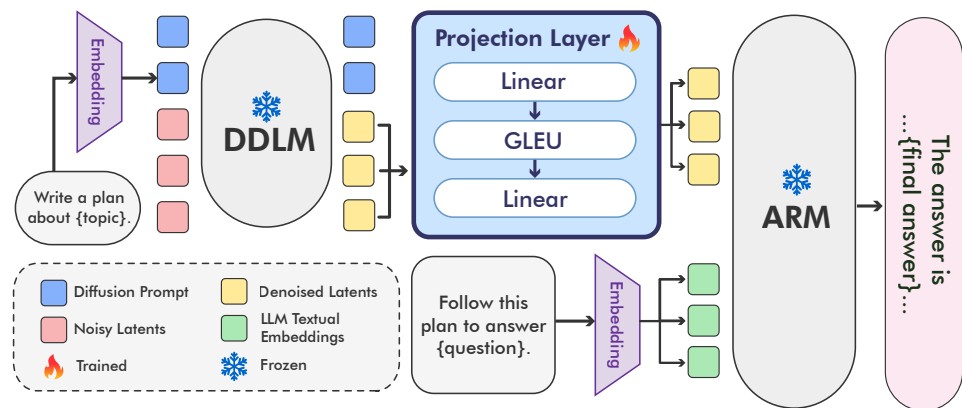

Figure 2: Overview of the latent-space collaboration pipeline. A discrete diffusion language model (DDLM) generates a structured plan from noisy latents. The plan is projected directly into the autoregressive model (ARM) embedding space through a learned projection layer (latent space). The ARM then conditions on the plan and the question to produce the final answer.

when necessary is worth exploring. This insight motivated the design of the following collaboration pathway.

**Latent-space collaboration.** Figure 2 shows the latent-space configuration, where a DDLM-based planner generates plans directly in the latent space, and an ARM-based executor produces the final answers. Communication between the models is enabled via a learned projection layer that maps DDLM states into the ARM executor's embedding space. The projector consists of a Linear–GELU–Linear stack trained to align DDLM plans with LLM embeddings. During inference, the DDLM latents are projected directly into the executor's hidden space, bypassing a raw textual plan. While this approach sacrifices interpretability, since the intermediate reasoning steps are no longer visible, it allows for richer and more expressive communication between models. In fact, recent work has shown that diffusion models can encode the correct answer in their latent representations before surface-level decoding (Li et al., 2025b), suggesting that latent exchange may unlock deeper reasoning signals that text-based communication cannot capture.

**End-to-End Pipeline** The reasoning pipeline begins with a question and a planning prompt. The prompt is embedded and processed by the DDLM, which iteratively denoises noisy latents into a structured plan representation. This plan can be communicated to the executor in two ways: as surface text appended to the question (text space) or as a latent embedding projected directly into the executor's representation space (latent space). The executor, instantiated as an ARM, then conditions on both the original question and the provided plan to generate the final answer.

## 4 COLLABORATION THROUGH TEXT SPACE

### 4.1 EXPERIMENTAL SETUP

This section details the experimental setup for the text-space collaboration presented in Section 3. To rigorously assess experimentally the benefits of collaboration between ARMs and DDLMs in this setting, we evaluate all the following possible combinations:

**ARM-only.** We first evaluate the performance of an ARM used in isolation, where it is directly prompted to answer the given questions without an explicit planning step.

**ARM → ARM.** Next, we evaluate a setup where one ARM acts as the planner, generating intermediate reasoning or guidance, and another instance of the same model serves as the executor, producing the final answer.

**DDLM-only and DDLM → DDLM.** In parallel, we first assess a DDLM in isolation as an executor-only baseline. We then evaluate a DDLM → DDLM setup, where two instances of the same DDLM are used, the first serving as planner and the second as executor.

**ARM → DDLM and DDLM → ARM.** Finally, we examine collaborative setups between ARMs and DDLMs, alternating their roles.

## 4.2 MODELS AND BENCHMARKS

**DDLMs.** We evaluate the setups described in Section 4.1 using two recently released masked diffusion models (MDMs) from 2025, LLada-8B-Instruct (Nie et al., 2025) and Dream-v0-Instruct-7B (Ye et al., 2025). We set the default sequence length to 256 tokens, providing the planner with enough capacity to generate reasoning plans while minimizing repetitions errors (see details in Appendix A.2).

**ARMs: Non-Reasoning Models.** Alongside the DDLMs, we consider two autoregressive models (ARMs) of comparable size, Qwen2.5-7B-Instruct(Yang et al., 2024) and Llama-3.1-8B-Instruct(Touvron et al., 2023; Dubey et al., 2024), for a fair comparison. We also include two smaller ARMs, Llama-3.2-3B-Instruct(Touvron et al., 2023; Dubey et al., 2024) and Qwen2.5-3B-Instruct(Yang et al., 2024), to illustrate the results with smaller models.

**ARMs: Reasoning Models.** We additionally compare the previously described collaboration setups against reasoning models to contextualize their performance relative to state-of-the-art systems. Specifically, we evaluate two reasoning-oriented models: Qwen3-1.7B (Yang et al., 2025a) and DeepSeek-R1-Distill-Qwen-7B (Yang et al., 2024). The latter is a distilled variant trained to compress the reasoning ability of the original DeepSeek-R1 into a smaller Qwen-7B backbone. For simplicity, we refer to this distilled variant as the *DeepSeek model* throughout the remainder of the paper, even though it is not the full original model. Our objective in these experiments is not to match the performance of specialized reasoning models, but to study the potential of hybrid ARM-DDLM collaborations among general language models when compared to strong state-of-the-art reasoning baselines.

**Benchmarks.** We evaluate on a diverse suite of reasoning benchmarks: ARC-E and ARC-C (Clark et al., 2018), science exam questions at Easy and Challenge difficulty; MMLU (Hendrycks et al., 2021b;a), spanning mathematics, history, computer science, and law; and AIME 2024 (Veeraboina, 2024; Jia, 2024), a high-school mathematics competition. We further include DART-1 through DART-5 (Tong et al., 2024), a large-scale mathematical reasoning benchmark covering five difficulty levels. These benchmarks are standard in the literature and together provide broad coverage of reasoning domains. For each evaluation, we use the greater of 200 samples or the full benchmark size.

## 4.3 RESULTS

Overall, text-space experiments, reported in Table 1, show three consistent patterns. **(i) Diffusion excels in isolation but struggles in dual roles.** Dream-v0-7B as a single DDLM achieves the best DART-1 score overall (83.5) and strong results on ARC-E (96.5) and MMLU (62.5), surpassing comparably sized ARMs such as Llama-3.1-8B (68.5 on DART-1, 63.5 on MMLU). However, chaining diffusion models reduces robustness: Dream-v0-7B in a DDLM→DDLM setup drops from 83.5 to 76.5 on DART-1 and from 46.5 to 44.0 on DART-3, while LLaDA-8B falls from 80.0 to 68.5 on DART-1, indicating compounding errors when both planner and executor are diffusion-based.

**(ii) ARM planning is steadier than diffusion planning, though gains are uneven.** For example, Qwen2.5-7B improves from 68.0 to 72.5 on DART-1 when adding an ARM planner, but loses 6 points on MMLU (61.0 vs. 67.0). Similarly, LLaDA-8B+Qwen2.5-7B with ARM→DDLM reaches 80.5 on DART-1, whereas reversing the roles (DDLM→ARM) drops performance to 73.5. Dream-v0-7B+Llama-3.1-8B shows an even sharper contrast: ARM→DDLM achieves 77.0 on DART-1, while DDLM→ARM collapses to 55.5. These trends suggest that ARM planners in text space produce more stable improvements, while diffusion planners introduce higher variance.

**(iii) Hybrids fall short of specialized reasoning models but establish useful baselines.** DeepSeek-R1, a reasoning-optimized ARM, achieves 85.5 on DART-3 and 28.5 on AIME, far above any hybrid. Qwen3-1.7B reaches 86.0 on DART-2 and 68.5 on MMLU, again stronger overall. Yet hybrids demonstrate non-trivial competence: LLaDA-8B+Llama-3.1-8B with ARM→DDLM scores 81.0 on DART-1, only 8 points below DeepSeek-R1 (89.0), despite using smaller general-purpose language models. Taken together, these results confirm that diffusion is a strong standalone

Table 1: Consolidated evaluation results on text-space collaboration across all model combinations and reasoning benchmarks.
† Evaluated with `enable_thinking=True`, ‡ with `enable_thinking=False`.

| Model / Combination | Setup | ARC-E | ARC-C | DART-1 | DART-2 | DART-3 | DART-4 | DART-5 | AIME | MMLU |
|---|---|---|---|---|---|---|---|---|---|---|
| Qwen2.5-3B | ARM | 91.0 | 86.5 | 58.0 | 34.0 | 29.5 | 17.5 | 11.5 | 1.0 | 61.0 |
| | ARM → ARM | 89.5 | 81.5 | 64.0 | 49.0 | 30.5 | 24.0 | 16.0 | 0.0 | 59.0 |
| Qwen2.5-7B | ARM | 94.5 | 90.5 | 68.0 | 48.5 | 35.0 | 35.0 | 19.5 | 2.5 | 67.0 |
| | ARM → ARM | 96.5 | 89.0 | 72.5 | 56.5 | 35.0 | 33.5 | 20.0 | 3.0 | 61.0 |
| Llama-3.2-3B | ARM | 87.5 | 79.5 | 44.5 | 35.5 | 28.0 | 34.5 | 36.5 | 3.5 | 54.0 |
| | ARM → ARM | 91.0 | 80.0 | 47.5 | 37.0 | 31.0 | 30.5 | 30.0 | 0.0 | 56.5 |
| Llama-3.1-8B | ARM | 87.5 | 79.5 | 68.5 | 50.5 | 36.0 | 36.0 | 20.5 | 2.5 | 63.5 |
| | ARM → ARM | 91.0 | 82.0 | 74.0 | 58.5 | 35.5 | 35.0 | 20.5 | 3.0 | 64.0 |
| LLaDA-8B | DDLM | 94.5 | 87.5 | 80.0 | 59.0 | 41.0 | 35.5 | 15.0 | 1.5 | 59.0 |
| | DDLM → DDLM | 93.0 | 83.5 | 68.5 | 53.5 | 45.0 | 32.5 | 17.0 | 1.0 | 52.0 |
| Dream-v0-7B | DDLM | 96.5 | 90.5 | 83.5 | 56.5 | 46.5 | 41.0 | 20.5 | 1.5 | 62.5 |
| | DDLM → DDLM | 95.0 | 91.5 | 76.5 | 59.0 | 44.0 | 38.5 | 19.0 | 3.0 | 65.5 |
| LLaDA-8B + Qwen2.5-3B | ARM → DDLM | 90.5 | 86.0 | 78.0 | 65.5 | 42.5 | 35.0 | 17.5 | 1.0 | 57.0 |
| | DDLM → ARM | 88.5 | 85.0 | 68.0 | 44.5 | 29.0 | 23.0 | 14.0 | 0.5 | 55.5 |
| LLaDA-8B + Qwen2.5-7B | ARM → DDLM | 95.0 | 91.0 | 80.5 | 61.5 | 48.0 | 31.5 | 18.5 | 1.0 | 61.5 |
| | DDLM → ARM | 92.5 | 88.5 | 73.5 | 58.0 | 37.5 | 37.0 | 16.0 | 1.5 | 54.0 |
| LLaDA-8B + Llama-3.2-3B | ARM → DDLM | 93.0 | 84.0 | 73.0 | 59.5 | 44.0 | 36.0 | 25.0 | 1.0 | 52.0 |
| | DDLM → ARM | 90.5 | 82.5 | 53.5 | 43.0 | 35.5 | 30.0 | 27.0 | 0.0 | 52.5 |
| LLaDA-8B + Llama-3.1-8B | ARM → DDLM | 91.5 | 86.5 | 81.0 | 62.0 | 48.0 | 32.0 | 20.0 | 1.0 | 60.5 |
| | DDLM → ARM | 91.5 | 82.5 | 74.0 | 60.0 | 38.5 | 37.0 | 16.5 | 1.5 | 56.5 |
| Dream-v0-7B + Qwen2.5-3B | ARM → DDLM | 92.0 | 87.0 | 79.0 | 59.0 | 47.0 | 34.5 | 17.0 | 2.0 | 61.5 |
| | DDLM → ARM | 92.0 | 86.0 | 65.5 | 40.5 | 31.0 | 25.0 | 12.5 | 0.5 | 60.5 |
| Dream-v0-7B + Qwen2.5-7B | ARM → DDLM | 96.0 | 92.0 | 76.0 | 61.0 | 48.0 | 40.0 | 17.5 | 2.5 | 63.5 |
| | DDLM → ARM | 95.5 | 88.5 | 60.5 | 42.0 | 34.0 | 28.5 | 16.0 | 1.0 | 64.5 |
| Dream-v0-7B + Llama-3.2-3B | ARM → DDLM | 95.0 | 89.5 | 74.5 | 60.0 | 48.0 | 40.5 | 23.5 | 3.0 | 60.5 |
| | DDLM → ARM | 91.0 | 79.0 | 54.0 | 39.5 | 32.0 | 31.5 | 27.5 | 0.0 | 59.5 |
| Dream-v0-7B + Llama-3.1-8B | ARM → DDLM | 96.5 | 92.0 | 77.0 | 61.0 | 47.5 | 41.0 | 16.0 | 2.0 | 62.5 |
| | DDLM → ARM | 94.0 | 86.5 | 55.5 | 33.0 | 37.0 | 28.0 | 26.0 | 0.5 | 64.0 |
| **Reasoning Models (Baselines)** | | | | | | | | | | |
| LLaDA-8B + DeepSeek-R1 | ARM | 94.0 | 88.0 | 89.0 | 81.5 | 85.5 | 75.0 | 61.5 | 28.5 | 60.0 |
| | DDLM → ARM | 92.5 | 88.5 | 73.5 | 58.0 | 37.5 | 37.0 | 16.0 | 1.5 | 54.0 |
| LLaDA-8B + Qwen3.1-7B | ARM† | 92.0 | 86.0 | 89.5 | 86.0 | 83.0 | 73.0 | 49.0 | 8.5 | 68.5 |
| | DDLM → ARM‡ | 91.0 | 82.5 | 87.0 | 70.0 | 58.5 | 46.0 | 27.0 | 1.0 | 52.5 |

planner but is fragile in stacked diffusion settings, that ARM planners yield steadier text-space performance, and that hybrid systems form a stepping stone toward latent-space collaboration.

## 5 Collaboration Through Latent Space

### 5.1 Diagnosing Failure Modes

To motivate latent-space collaboration, we perform a diagnostic analysis of text-space outputs to determine whether the DDLM → ARM setup could benefit from latent-space intervention. We focus on two specific subsampled setups: **(i) Setup X**: Questions where DDLM → ARM fails due to planning errors (the DDLM), i.e., cases where the ARM could have succeeded given a more coherent plan. **(ii) Setup Y**: Questions where DDLM → DDLM succeeds but DDLM → ARM fails, indicating that the executor (ARM) is the limiting factor. This selection procedure is illustrated in Figure 4. We then define diagnostic percentages to quantify planner versus executor contributions:

$$\textbf{Percentage}_i = \frac{\#\{\text{Setup } i \text{ samples}\}}{\#\{\text{Incorrect samples in DDLM} \rightarrow \text{ARM}\}} \quad \text{for } i \in \{X, Y\}.$$

Table 2: Evaluation of DeepSeek-R1-Distill-Qwen-7B and Qwen3-1.7B on reasoning benchmarks, including text-space vs. latent-space collaboration.

| Model / Setting | ARC-E | ARC-C | DART-1 | DART-2 | DART-3 | DART-4 | DART-5 | AIME24 | MMLU |
|---|---|---|---|---|---|---|---|---|---|
| DeepSeek-R1 (ARM) | 94.0 | 88.0 | 89.0 | 81.5 | 85.5 | 75.0 | 61.5 | 28.5 | 60.0 |
| Avg. tokens | 398 | 504 | 1420 | 1833 | 2076 | 2418 | 3068 | 3832 | 760 |
| Qwen3-1.7B (ARM) | 92.0 | 86.0 | 89.5 | 86.0 | 83.0 | 73.0 | 49.0 | 8.5 | 68.5 |
| Avg. tokens | 282 | 397 | 1024 | 1669 | 2112 | 2550 | 3106 | 4036 | 984 |
| **Baselines** | | | | | | | | | |
| Llama-3.1-3B (ARM only) | 87.5 | 79.5 | 44.5 | 35.5 | 28.0 | 34.5 | 36.5 | 3.5 | 54.0 |
| Avg. tokens | 4 | 4 | 17 | 16 | 22 | 28 | 41 | 462 | 4 |
| LLaDA-8B + Llama-3.1-3B (Text) | 90.5 | 82.5 | 53.5 | 43.0 | 35.5 | 30.0 | 27.0 | 0.0 | 52.5 |
| Avg. tokens | 4 | 4 | 10 | 14 | 16 | 12 | 20 | 503 | 4 |
| **Latent-space Collaboration** | | | | | | | | | |
| 64 tokens plan | 85.0 | 78.5 | 78.5 | 62.5 | 57.0 | 63.0 | 54.0 | 12.5 | 52.0 |
| 128 tokens plan | 87.5 | 76.5 | 70.5 | 43.0 | 43.0 | 49.0 | 36.0 | 14.0 | 44.0 |
| 256 tokens plan | 85.0 | 81.0 | 70.0 | 62.0 | 50.0 | 62.0 | 52.5 | 14.0 | 15.5 |
| Avg. output tokens | 2 | 2 | 4 | 5 | 5 | 5 | 6 | 14 | 2 |

These ratios measure the relative impact of planning versus execution errors in the collaboration pipeline. The performance comparison between Setup X (planner-focused) and Setup Y (executor-focused) across various benchmarks is presented in Figure 3. The results reveal that, under text-space collaboration (DDLM→ARM), the majority of percentage errors originate from the planner rather than the executor, indicating that the planner contributes more significantly to performance degradation. In contrast, under latent-space collaboration, we observe a shift: the executor becomes the primary source of errors. This suggests that improvements achieved through latent-space collaboration have primarily benefited the planner component. We provide the exact percentages in A.1.

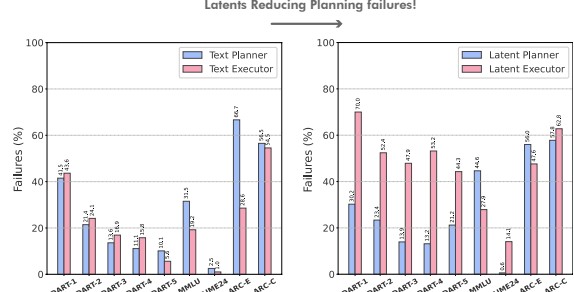

Figure 3: **Planner vs. executor failures in text- vs. latent-space collaboration.** Results for LLaDA-8B-Instruct and Llama-3.1-3B-Instruct. Latent-space collaboration substantially reduces planning errors compared to text-space.

## 5.2 EXPERIMENTAL SETUP

To enable DDLM → ARM collaboration, we train a Linear-GELU-Linear projector (Figure 2) that maps DDLM hidden states (latents) into the ARM embedding space. We construct the training data by generating DDLM latent representations from 35K samples, drawn uniformly (5K each) from ARC_Easy, ARC_Challenge, DART1, DART2, DART3, DART4, and DART5. For each sample, we produce latents with fixed output lengths of 64, 128 and 256 tokens. The projector is trained with both ARM and DDLM weights frozen. Training uses a cross-entropy objective, comparing ARM predictions, conditioned on latent-space inputs, with the corresponding ground-truth tokens from the benchmarks. In our setup, we use LLaDa-8B-Instruct as the DDLM planner and Llama-3.2-3B-Instruct as the ARM executor.

## 5.3 RESULTS

**Latent-space vs Text-space Collaboration** The results of the latent-space collaboration compared to the text-space baseline are reported in Table 2 and illustrated in Figure 5. While performance on ARC-E (85.0 vs. 90.5) and ARC-C (81.0 vs. 82.5) remains comparable across the two settings, the latent space consistently yields substantially higher accuracy on the DART benchmarks: e.g., DART-1 (78.5 vs. 53.5), DART-2 (62.5 vs. 43.0), DART-3 (57.0 vs. 35.5), DART-4 (63.0 vs. 30.0), and DART-5 (54.0 vs. 27.0). On AIME, the latent approach performance reaches 12.5% with the projector trained on 64 tokens and 14% with the projector trained on 128 or 256 tokens, compared to 0.0% for the text space. Significantly, this improvement is obtained even though the projector between DDLM and ARM was trained without using any data from AIME or MMLU,

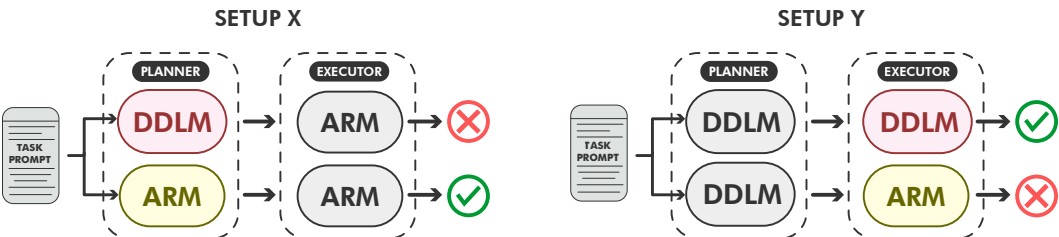

Figure 4: Diagnostic configurations for attributing errors to planner or executor. **Setup X** tests whether failures stem from the planner: if replacing the diffusion planner (DDLM) with an autoregressive planner (ARM) fixes the output, the error is attributed to the DDLM. **Setup Y** tests executor reliability: if a diffusion executor succeeds where an ARM executor fails, the limitation lies in the executor.

yet it still generalizes to deliver markedly stronger results on these challenging evaluations. This highlights the promise of latent-space communication as an effective channel for planner–executor collaboration.

One possible explanation for why the setup with 64 latent tokens performs better on average than the other configurations is that it introduces less redundancy. In fact, as shown in Appendix Table 4, the number of repetitions in diffusion with 64 tokens in LLADA is comparable to that of Qwen2.5-7B and remains similarly low.

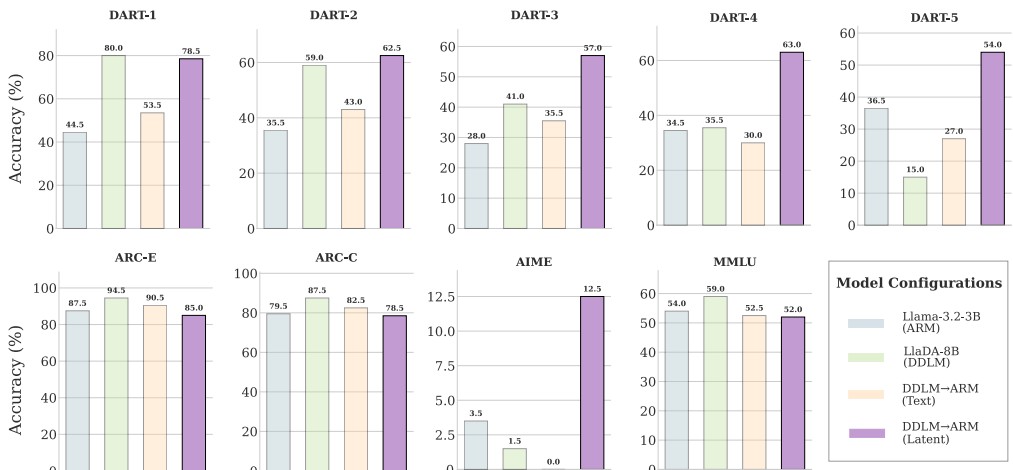

Figure 5: **Benchmark comparison of text-space vs. latent-space collaboration.** Accuracy of isolated models (LLaMA-3.2-3B ARM, LLaDA-8B DDLM) and collaborative configurations. In the latent setting, DDLM (64-token planner) combined with the ARM executor consistently outperforms text-space collaboration on DART and AIME, while maintaining comparable performance on ARC and MMLU.

**Token Budget and Efficiency** We explicitly control the length of diffusion-generated plans (64, 128, 256 tokens). Results demonstrate that longer is not always better: surprisingly, the 64-token setup achieves the best overall trade-off on average (Table 2). Most importantly, latent space collaboration is markedly more efficient than both baselines (DDLM→ARM in text space and ARM-only) as well as reasoning models. With just 64 planner tokens and an average of 5 executor tokens, it surpasses Qwen3 on DART-5 while using only **2.2% tokens**, and outperforms it on AIME with only **1.9% tokens**. Although it does not yet match DeepSeek-R1 in raw accuracy, it achieves competitive performance at a fraction of the cost, as reflected by the average token usage in Table 2.

# 6 DISCUSSION

Our investigation provides several insights into the collaboration between discrete diffusion language models (DDLMs) and autoregressive models (ARMs) under a planner–executor framework.

First, the **communication channel** between planner and executor strongly shapes performance. Text-space collaboration is simple and interpretable, but constrained by the limitations of diffusion outputs. Latent-space collaboration mitigates this bottleneck, enabling the executor to leverage diffusion's internal representations directly. This shift produces substantial accuracy gains on challenging benchmarks such as DART and AIME, where text-based prompting alone struggles. Interestingly, latent exchange also alters the error landscape: planner-driven failures dominate in text space, whereas executor errors become more prominent in latent space, indicating a redistribution in the sources of error.

Second, **efficiency** emerges as a natural outcome of hybrid collaboration. Latent-space setups achieve competitive or superior accuracy to strong reasoning baselines while using one to two orders of magnitude fewer tokens. For example, with only 64 planner tokens plus ∼5 executor tokens, our system surpasses Qwen3 on DART-5 and AIME, despite requiring only 2.2% and 1.9% tokens respectively. This efficiency indicates that hybridization does not merely replicate autoregressive reasoning with fewer steps, but rather restructures the compute–accuracy trade-off in a qualitatively manner.

Finally, our results highlight several **open challenges**. While latent communication improves robustness, it sacrifices interpretability, raising questions about how to inspect or align internal reasoning signals. Moreover, diffusion planners sometimes overproduce redundant steps, suggesting a need for adaptive mechanisms that balance global revision with conciseness. Future work may also explore scaling effects, alternate projection architectures, or training regimes that align planner and executor representations more tightly.

# 7 CONCLUSION

We presented a systematic study of planner–executor collaboration between discrete diffusion and autoregressive models for reasoning tasks. By comparing multiple setups and communication channels, we found that positioning DDLMs as planners and ARMs as executors, coupled with latent-space exchange, offers the most effective balance of accuracy, efficiency, and robustness. This hybrid approach achieves substantial improvements on challenging mathematical benchmarks while drastically reducing token budgets, demonstrating a new path for budget-aware reasoning.

Beyond empirical gains, our work shows that hybrid reasoning is not just a compromise between paradigms but an opportunity to design a division of labor explicitly across compute, fluency, and robustness.

Looking ahead, several directions remain open. Improving interpretability in latent communication, aligning planner and executor through joint training, and extending hybrid frameworks to multimodal or agentic settings are promising avenues. We see this work as a foundation for future systems that integrate diverse generation mechanisms into modular reasoning pipelines, enabling both more efficient and more interpretable large-scale reasoning.

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

# A  APPENDIX

## A.1  PERFORMANCE COMPARISON OF PLANNER VS. EXECUTOR ISSUES

Table 3: Performance comparison of planner vs. executor issues for LLaDA-8B-Instruct and Llama-3.1-3B-Instruct under **Text-Space** vs. **Latent-Space** collaboration.

| Benchmark | Planning Failures (%) | Execution Failures (%) | Error Gap (%) |
|---|---|---|---|
| **LLaDA-8B + Llama-3.1-3B (Text-Space Pipeline)** | | | |
| DART-1 | 41.50 | **43.64** | 2.14 |
| DART-2 | 21.43 | **24.14** | 2.71 |
| DART-3 | 13.60 | **16.92** | 3.32 |
| DART-4 | 11.11 | **15.79** | 4.68 |
| DART-5 | **10.12** | 5.63 | 4.49 |
| MMLU | **31.52** | 19.23 | 12.29 |
| AIME24 | **2.54** | 1.03 | 1.51 |
| ARC-E | **66.67** | 28.57 | 38.10 |
| ARC-C | **56.52** | 54.55 | 1.97 |
| **LLaDA-8B + Llama-3.1-3B (Latent-Space Pipeline)** | | | |
| DART-1 | 30.23 | **70.00** | 39.77 |
| DART-2 | 23.37 | **52.42** | 29.05 |
| DART-3 | 13.95 | **47.88** | 33.93 |
| DART-4 | 13.15 | **53.19** | 40.04 |
| DART-5 | 21.21 | **44.29** | 23.08 |
| MMLU | **44.64** | 27.90 | 16.74 |
| AIME24 | 0.58 | **14.07** | 13.49 |
| ARC-E | **56.00** | 47.61 | 8.39 |
| ARC-C | 57.80 | **62.79** | 4.99 |

## A.2  MORE INSIGHTS ON PLANNER REPETITION OF TOKENS

We will perform a qualitative assessment to identify prompt repetition errors in the planner text in the setup DDLM $\rightarrow$ ARM in the text space, using 256, 128 and 64 tokens for the LlaDa-8B-Instruct model, as well as Qwen2.5-7B-Instruct and Dream-v0-7B-Instruct for comparison. We use the following metrics proposed in (Aoki et al., 2023):

- Distinct-3 (D-3)
- Repetition-4 (R-4)
- Lexical Repetition (LR-n)

**Distinct-3 (D-3)** calculates the percentage of unique 3-grams over all 3-grams. The value of Distinct-3 takes values between 0 and 1, with the closer to 1 indicating that the text is more diverse at the 3-gram level. Let $D_3$ be the number of unique 3-grams in the text and $T_3$ be the total number of 3-grams in the text. Distinct-3 is then computed by the following formula:

$$\text{Distinct-3} = \frac{D_3}{T_3} \times 100$$

**Repetition-4(R-4)** Let $T$ be the total number of sentences in the text, $R_t$ be the number of 4-grams repeated in a sentence $t$, and $I(x)$ be an indicator function (1 if $x$ is true, 0 if $x$ is false). Then Repetition-4 is calculated as follows:

$$\text{Repetition-4} = \frac{1}{T} \sum_{t=1}^{T} I(R_t > 1) \times 100$$

**Lexical Repetition (LR-n)** computes the average percentage of 4-grams that occur at least $n$ times in the generated text. Let $G$ be the total number of possible 4-grams in all texts and $L_g$ be the number of repetitions of $G$, then Lexical Repetition (LR-n) is calculated by the following formula:

$$\text{Lexical Repetition} = \frac{1}{G} \sum_{g=1}^{G} I(L_g \geq n) \times 100$$

Table 4: Repetition Evaluation

| Simulation | D-3 | R-4 | LR-n |
|---|---|---|---|
| | ↑ | ↓ | ↓ |
| Qwen2.5-7B-Instruct (Baseline) | 98.47 | 3.05 | 0.64 |
| LLaDA-8B-Instruct (256 tokens) | 83.05 | 10.52 | 6.74 |
| LLaDA-8B-Instruct (128 tokens) | 93.15 | 5.33 | 3.43 |
| LLaDA-8B-Instruct (64 tokens) | 96.85 | 3.37 | 1.33 |
| Dream-v0-7B-Instruct (256 tokens) | 62.02 | 3.26 | 2.15 |

**Results** In table 4, the ↑ indicates that a larger value corresponds to better performance, while the ↓ indicates that a smaller value corresponds to better performance.

We observe a tendency for greater repetition in the plans generated by LLADA as the number of tokens increases.The number of repetitions in diffusion with 64 tokens in LLADA is comparable to that of Qwen2.5-7B and remains similarly low.

### A.3 PROMPTS FOR LLM EXPERIMENTS

### A.4 PLANNER PROMPT

```
You are a careful problem-solving planner.

Task: Produce ONLY a short list of HINTS that help solve the question.
Do NOT state or imply the final answer. Do NOT mention any option letter
(A, B, C, or D). Do NOT quote any option text verbatim.
If you find yourself about to reveal a specific option or an answer,
replace it with "[HIDDEN]".

Format:
- Key facts to recall (2-4 bullets)
- Reasoning steps or elimination rules (2-5 bullets)
- Useful equations or definitions (if relevant)
- Edge cases or common traps (optional)

Be concise (<=120 words). No "Answer:" line. No letters A-D.

Question (stem only):
{question}
```

### A.5 EXECUTOR PROMPT

```
You are an expert in solving multiple-choice questions.
Given the following plan or reasoning, please solve the question.
If the plan contains any explicit answer or option letter, ignore it and
solve from the hints + question only.
```

```
Plan:
{plan}
{question}
```

