# OpenReview forum: "Planner and Executor: Collaboration between Discrete Diffusion And Auto-regressive Models in Reasoning"
_ICLR.cc/2026/Conference — ICLR 2026 Conference Withdrawn Submission_

### Official Review · Reviewer_54Fx · 2025-10-30

**Soundness:** 2
**Presentation:** 1
**Contribution:** 2
**Rating:** 2
**Confidence:** 4

**Summary:**

This paper investigates a hybrid "planner-executor" framework for reasoning tasks, combining a Discrete Diffusion Language Model (DDLM) as the planner and an Autoregressive Model (ARM) as the executor. The core contribution is the exploration of two collaboration channels: text-space and latent-space, with the latter using a learned projector to map DDLM states into the ARM's embedding space. The central claim is that this hybrid approach, particularly with latent-space communication, achieves competitive reasoning accuracy with a drastically reduced token budget.

**Strengths:**

- The "DDLM-ARM" cooperation framework is worth studying.
- I agree that DDLM has advantages on token budget control, providing a more controlable thinking process.
- The study provides a systematic comparison of different planner-executor pairings (ARM→ARM, DDLM→DDLM, ARM→DDLM, DDLM→ARM) and collaboration spaces (text vs. latent).

**Weaknesses:**

- Usually, reasoning algorithms require experiments on MATH / Code generation or common reasoning tasks like GPQA. This paper lacks most of these benchmarks. For AIME, it is doubtful that all these models only achieve near zero on AIME. The metrics are too low.
- Moreover, Tab. 1 demonstrates a large degradation in all evaluated metrics. Make the effectiveness of the "planner-executor" framework doubtful.
- Actually, I do not think text space cooperation could do harm to the performance as in Tab. 1. By writing a better prompt, it should be at least the same performance without collaboration.
- As to latent space collaboration, there lacks a comparison between using ARM and DDLM as the planner.
- Moreover, the authors do not provide any specific examples to see what plan and answer will the models generate.
- Layout and grammar mistakes (like line changes in L027 & L055)

**Questions:**

1. If we choose latent space, how many times will the DDLM conduct the forward pass? I suppose it to be only once. If only once, AR model can also be used with fixed token budget and high-level semantics. What is its advantage over AR model.
2. Have the authors tried to finetune this collaboration framework with RL for reasoning?

---

### Official Review · Reviewer_nR4U · 2025-10-31

**Soundness:** 1
**Presentation:** 1
**Contribution:** 2
**Rating:** 2
**Confidence:** 5

**Summary:**

This paper addresses the high computational cost (i.e., token count) of reasoning in high-performing Autoregressive Models (ARMs), which typically rely on long chain-of-thought sequences. It proposes a hybrid planner-executor framework that combines the strengths of ARMs with those of Discrete Diffusion Language Models (DDLMs), which can generate content in parallel with a fixed token budget.

**Strengths:**

- The primary strength is the highly effective proposal of a latent-space DDLM $\rightarrow$ ARM collaboration. This work moves beyond just using diffusion as another text generator and instead leverages its internal representations for planning, which is a significant conceptual leap. The resulting improvement in token efficiency for complex reasoning is a major finding.
- The systematic comparison of all four planner-executor pairings and both communication channels (text vs. latent) provides a complete picture. The diagnostic failure analysis (Sec 5.1) is a key strength, providing a data-driven justification for the paper's core hypothesis.
- The gains are not marginal. The leap from 0.0% to 14.0% on AIME and 27.0% to 54.0% on DART-5 by switching from text to latent space is dramatic. Beating a strong baseline (Qwen3) on these tasks with ~98% fewer tokens is a highly compelling result

**Weaknesses:**

- The major weakness of the paper is the lack of novelty. Simply plugging the Diffusion Models to the ARMs without further analysis and algorithm design is weak for an academic paper at ICLR.
- The details of the alignment between the Diffusion Models and the ARM are omitted. For instance, how do you configure the Diffusion Models? How do you determine the time step to extract the latent representation or text output? It introduced lots of confusion here. The text output is obtained from the final hidden representation after an embedding map. Do you use the final hidden representation as the latent space input for ARM or intermediate hidden representation?
- The experiment settings are insufficient and unorganized. The experiments demonstrated in Table 1 fail to clearly present the comparison between different settings. Please reconsider your choice and determine the critical data which can support your argument.

**Questions:**

See Weaknesses.

---

### Official Review · Reviewer_gaaM · 2025-11-01

**Soundness:** 1
**Presentation:** 1
**Contribution:** 2
**Rating:** 0
**Confidence:** 4

**Summary:**

This work explores developing hybrid frameworks that combine both discrete diffusion language models (DDLM) and autoregressive language models (ARM) for reasoning tasks. Broadly they consider pipelined approaches where one model acts as the “planner” and the other model acts as the “executor”. They present results for multiple choice and math benchmarks across various instantiations of this general framework, integrating both token plans and latent plans (e.g. final layer features) with a trainable projection layer.

**Strengths:**

This paper presents an interesting approach for combining the strengths of two different generation paradigms: diffusion LMs and AR LLMs.

The authors present a comprehensive sweep over instantiations of their planner/executor framework. They analyze using diffusion LMs and AR LMs for either role and additionally explore using latent plans instead of raw textual outputs.

The diagnostic analysis is a smart experimental setup to probe the behavior of different configurations and identify the source of different errors.

**Weaknesses:**

I don’t think the motivation for combining DLMs and AR LMs is very convincing. It is widely accepted that introducing some textual reasoning before answering can be more effective than answering directly (chain-of-thought, R1, etc.). I don’t think the benefit of using a diffusion model to generate the plan then generating the answer with an autoregressive model is very clear.

The experimental results in Table 2 are generally mixed and are very difficult to parse. It presents 100+ results and the organization makes it challenging to compare the collaboration results with their base models, etc.. Without some kind of aggregation, it is very difficult to observe trends by just inspection. It does not look like the collaboration consistently outperforms both constituent models. Or the simpler setting of using one model to serve as both roles.

While the latent plan idea is interesting, it is also SFT-ing directly on the target task while all other results are 0-shot. It can essentially act as a form of prefix-tuning with ground truth labels which would likely lead to downstream performance improvements regardless of the addition of the “plan”. I suspect the gold supervision is a strong contributor to the improvement in that setting.

While efficiency is discussed in terms of the number of tokens, the discussion is very incomplete. Discrete diffusion models are generally significantly slower than corresponding autoregressive models of the same scale due to the lack of kv-caching, so the token budget paints an incomplete picture. Furthermore, just focusing on tokens for the executor is incomplete. Generating the plan also has overhead.

**Questions:**

What is the inference overhead of various configurations?

---

### Official Review · Reviewer_ChMx · 2025-11-02

**Soundness:** 2
**Presentation:** 2
**Contribution:** 2
**Rating:** 2
**Confidence:** 5

**Summary:**

This paper explores a hybrid reasoning framework that combines Discrete Diffusion Language Models (DDLMs) and Autoregressive Models (ARMs). The authors systematically study four planner–executor configurations (DDLM and ARM, text vs. latent collaboration) and find that DDLM to ARM collaboration in latent space achieves the best performance. Experiments show large gains on reasoning benchmarks like DART-5 and AIME24, demonstrating that latent-space communication can provide both efficiency and accuracy.

**Strengths:**

1. Provides a systematic exploration of planner–executor combinations between DDLMs and ARMs, covering both text-space and latent-space collaboration.

2. Demonstrates that latent-space communication enables more efficient and accurate reasoning, offering a practical way to combine diffusion’s global planning with autoregression’s step-by-step precision.

**Weaknesses:**

1. The comparison may not be fully fair: while DDLM-generated planning is performed zero-shot, the latent-space variant involves additional training. Although the authors claim no overlap with benchmark datasets (e.g., MMLU, DART), the training data are all math-related, making it unclear whether the observed AIME improvements stem from training effects rather than the latent vs. discrete planning difference. Hence, the conclusion in Figure 3 is not fully supported.

2. Section 4.3 concludes that using AR as planner and DDLM as executor yields better results, but Section 5 shifts focus to DDLM as planner without explaining the motivation or rationale for this change—this weakens the conceptual coherence of the study.

**Questions:**

1. Some line breaks in the paper are inconsistent (e.g., line 026 and line 054).
2. The paper lacks training details (e.g., GPU type, batch size, learning rate, training duration), which makes it difficult for others to reproduce the results.

---

### Note · Authors · 2026-01-24

I have read and agree with the venue's withdrawal policy on behalf of myself and my co-authors.